# Digital Storytelling and Community Engagement to Find Missing TB Cases in Rural Nuh, India

**DOI:** 10.3390/tropicalmed7030049

**Published:** 2022-03-11

**Authors:** Subhi Quraishi, Hilmi Quraishi, Hemlata Yadav, Ayushi Singh, Ilmana Fasih, Nathaly Aguilera Vasquez, Lavanya Huria, Tripti Pande, Olive Mumba, Vishnu Vardhan Kamineni, Amera Khan

**Affiliations:** 1ZMQ Development, New Delhi 110034, India; hilmi@zmq.in (H.Q.); ayushi@zmq.in (A.S.); ilmana@zmq.in (I.F.); 2McGill International TB Centre, Research Institute of the McGill University Health Center, Montreal, QC H4A 3S5, Canada; nathaly.aguileravasquez@mail.mcgill.ca (N.A.V.); lavanya.huria@mail.mcgill.ca (L.H.); 3Independent Researcher, Washington, DC 20001, USA; tripti.pande@hotmail.com; 4The Global Fund to Fight AIDS, Tuberculosis and Malaria, 1218 Geneva, Switzerland; olive.Mumba@theglobalfund.org; 5TB REACH External Consultant M&E Team, 1218 Geneva, Switzerland; vvkamineni@gmail.com; 6Stop TB Partnership, 1218 Geneva, Switzerland; amerak@stoptb.org

**Keywords:** tuberculosis, active case finding, digital storytelling, community engagement

## Abstract

Nuh, Haryana, is one of India’s least developed districts. To improve TB case notifications, ZMQ carried out an active case-finding (ACF) intervention conducted by community health workers (MIRAs) using a digital TB storytelling platform to create TB awareness in the community. The combined storytelling and ACF intervention were conducted house-to-house or in community group settings. Steps included (A) the development of digital TB awareness-raising stories using a participatory approach called Story Labs; (B) the implementation of the intervention; and (C) process, outcome, and impact evaluation of these activities. Six digital stories were created and used during ACF in which 19,345 people were screened and 255 people were diagnosed with TB. Of 731 participants surveyed, the stories were well received and resulted in an increase in TB knowledge. ACF activities resulted in a 56% increase in bacteriologically confirmed TB and an 8% decrease in all forms of TB compared to baseline. All form notifications may have been impacted by COVID-19 lockdowns. Digital TB storytelling can improve TB awareness and knowledge, particularly for low-literacy populations. The use of these tools may benefit ACF campaigns and improve TB case finding.

## 1. Introduction

India has the highest tuberculosis (TB) burden in the world, reporting an estimated 2.6 million people with the disease in 2020 and representing 26% of the global TB burden. India faces challenges with underreporting and underdiagnosis of TB cases and accounts for 24% of the global gap between estimated TB cases and cases reported to national TB programs (NTPs). In the past two years, disruptions related to the COVID-19 pandemic had reversed some of the gains India had made in TB case detection. From 2019 to 2020, India contributed to 41% of the global drop in TB case notifications [1]. A person suffering from TB, if missed and left undiagnosed and untreated, can infect 5–15 individuals in close contact over the course of one year. TB thus affects the individuals, their families, and also society, directly or indirectly. Measures to find these “missing TB cases”, who are infectious but not presenting to the health care facilities, is the need of the hour, to curtail the further spread.

There are two key types of case-finding approaches: active and passive case-finding. The passive case-finding (PCF) method has its disadvantages as there may be delays both from the patient and the health system. Such delays may miss the diagnosis or result in delayed diagnosis, leading to increased risk of suffering, death, and catastrophic economic consequences for the patient. These missed opportunities play a crucial role in disease spread as it leads to longer periods of transmission, especially in vulnerable populations with poor living and working conditions [2]. The active case-finding approach on the other hand is systematic screening and clinical evaluation for TB in individuals who are at high risk of developing TB. A systematic review had been done on active case finding for TB in India [3]. Where the stated challenges are included poor awareness among the population and accessing the hard-to-reach communities. The aim of ACF is to diagnose people with undetected TB in the community; house-to-house screening for TB symptoms is insufficient without capacity building and behavior change communication of high-risk communities [3]. Along with awareness-raising, stigma reduction, and treatment support activities, community-based TB case finding can achieve early diagnosis, improved treatment outcomes, and reduced transmission. Furthermore, community involvement in TB care and prevention has proven to be effective and cost-saving in multiple areas of the TB care cascade. As a consequence, community-based case finding that integrates a solid community awareness and mobilization campaign along with a robust case-finding approach can have an impact on TB incidence, prevalence, and mortality.

Haryana, a state situated in northern India, has an annual TB case notification rate of 230 per 100,000 [4]. It comprises 22 districts where Nuh (formerly known as Mewat) is located. Nuh has a population of 1.01 million and is 98% rural with a literacy rate of only 38% in females and 56% in total [5]. Its inhabitants are primarily Meos, who are agriculturalists [6].

In 2018, the planning commission of the Government of India, NITI Aayog, established a list of 115 most aspirational districts (ADs) in India [7]. These districts are categorized based on 49 indicators from five identified thematic areas: health and nutrition, education, agriculture and water resources, financial inclusion and skill development, and basic infrastructure. In terms of the health indicators, Nuh is one of these ADs and has the lowest TB indicators, especially among rural women in India. In 2018, 3047 TB cases were notified in Nuh, representing only 4.6% of Haryana’s total TB notifications [4]. While the treatment completion rate is 85% in the public sector, it is only 2% in the private sector and is 16% lower among women than in men [4].

Female community health workers (known as accredited social health activists (ASHAs)) are often the ones to deliver primary care services to the population of Nuh. However, due to limited communication tools, they are often unable to deliver consistent health information in the community, which may lead to lower awareness and poor health indicators. The low literacy rates also limit capacity-building opportunities for using behavior change communication tools within the community. To address this problem, ZMQ Development (ZMQ), an organization based in Delhi, India, engaged and trained a group of female community health workers called Mobile Integrated Resources for All Women Needs (MIRAs) who use digital tools for consistent health communication, health tracking, and linking community to healthcare services in Nuh district.

ZMQ was awarded a TB REACH grant for 2020–2021 to implement a TB active case-finding (ACF) intervention among the population in Punhana block located in Nuh district to increase TB case detection whilst using storytelling to increase understanding and awareness of TB. Digital storytelling is a powerful tool that has been incorporated into the MIRA toolkit to engage communities with low literacy [8,9]. ZMQ has over two years of experience developing digital stories on various public health issues in close collaboration with community members, especially local women, as well as subject matter experts. The stories are created using a participatory approach, and the characters, voice-over, and scenarios are highly localized to further increase the engagement of the community. However, the digital storytelling interventions have not previously been formally evaluated. We conducted an evaluation to assess the effectiveness of the digital storytelling approach combined with ACF activities to increase TB awareness and subsequently increase TB notifications in the Punhana block of Nuh. The objectives of this project are to develop TB-related story content using “Story Labs” using community knowledge and experiences; use the digital storytelling for TB awareness and education; and assess the effectiveness of the storytelling in reaching its audience, increasing TB awareness, and subsequently increasing TB notifications.

## 2. Materials and Methods

### 2.1. Intervention Setting

ZMQ implemented an ACF and digital storytelling intervention between January 2020 and June 2021 in the Punhana block community in Nuh, Haryana. The combined TB storytelling and ACF intervention included the following steps: (A) the development of digital TB awareness-raising stories, (B) the implementation of ACF with the storytelling, and (C) the evaluation of these activities. The methods for each of these steps are further described below. To assess the intervention, a pre–post study design was used with purposive sampling.

### 2.2. A. Development of Digital TB Stories for the Your Story Teller Application—Story Labs

To create the TB awareness-raising stories, ZMQ used a community participatory approach and followed a systematic health communication process that consisted of developing and field-testing story concepts, messages, and format [10].

ZMQ has an open-source mobile-based digital application, called Your Story Teller (YST). YST houses digital stories to deliver health messages to communities. ZMQ has previously deployed YST to disseminate health messages on childhood pneumonia, menstrual hygiene, and immunization. Key features of the stories on YST include the participation of the community members in developing messages through participatory sessions called “Story Labs” and the use of animations featuring locally relevant characters, background information, and language.

The Story Labs rely on stakeholder involvement and are highly iterative. For this project, individuals visiting TB centers in Punhana block, local leaders, women’s self-help groups, local ASHA workers, and other local health workers were invited to voluntarily partake in the Story Labs. The Story Labs sessions included 20–25 individuals (adult women and men (>18 years of age) as well as adolescent girls (12–17 years of age)). While no monetary incentives were offered, refreshments were provided. ZMQ project staff facilitated these discussions within community health centers using a discussion guide. Topics such as TB screening, diagnosis, adherence to treatment, treatment completion, and myths and misconceptions around TB were covered. For each session, note-takers recorded participant details and key discussion points. The information collected from the Story Labs was then used to develop stories that delivered TB health messages tailored to the needs and questions of the community. The stories were subsequently animated and digitized into the YST application.

Technical aspects of the YST application’s user interface were assessed by health center staff. The design, animation, and voice-over for the stories were also field-tested with community members. A paper-based tool was used to document health center staff and community members’ comments, suggestions, and feedback on the usability and content of the stories. Once the feedback was incorporated, the stories were once again assessed for technical accuracy, design, and language to produce the final versions. This process resulted in a series of six stories, 5 to 7 min long, in the local Mewati dialect. Each story discussed one of the following topics: causes of TB, drug-resistant TB, prevention, signs and symptoms, stigma and discrimination, and TB treatment (see Appendix A). The stories were deployed on the YST app and downloaded onto smartphones used by MIRAs during their dissemination and screening activities. TB knowledge pre- and post assessments containing five different questions were also developed for each story and were included on the YST app to assess if the stories resulted in knowledge gain.

### 2.3. B. Implementation of Active Case Finding with Storytelling

An active case-finding (ACF) intervention, spearheaded by 50 trained MIRAs, incorporating two distinct screening approaches, (1) house-to-house dissemination and screening and (2) community-based dissemination and screening, was conducted from January 2020 to June 2021 in the Punhana block, Nuh district (hereafter evaluation population (EP)). The MIRA workers used mobile devices that included data collection tools for conducting ACF (i.e., screening, sputum collection, and referral forms) as well as the YST application which included a TB knowledge assessment and the stories developed for TB education and awareness.

For the house-to-house screening, MIRA workers are supposed to conduct house visits for identifying new pregnancies and new births and weekly visits to existing pregnant women to share information about services related to maternal and child health using the MIRA channel. During this interaction, MIRA also shares information about TB (common symptoms, transmission, diagnosis, and treatment) with the women and their associated families to conduct TB screening.

Each MIRA worker covered approximately 400 households in their respective village where they conducted MCH activities along with TB. As the MIRA worker engaged with the participants during the house-to-house visits, she determined which story was most appropriate and necessary based on the participant’s TB knowledge level and interest and subsequently chose which story to play on a case-by-case basis. MIRA screened approximately 190 to 210 houses based on family history, physical signs, or contact with a TB-infected person.

For the second ACF strategy (community screening), ZMQ volunteers engaged with local influencers and village leaders to plan community sessions to be held in conjunction with other prescheduled immunization drives, at school assemblies, or at Anganwadi centers (childcare centers). MIRA workers conducted the community TB awareness sessions and determined which story would be most beneficial for the group participants by assessing their level of knowledge and what themes they were interested in. Symptomatic screening following the same protocol as the house-to-house screening approach was conducted with community attendees after the storytelling sessions.

If an individual was identified as having TB signs or symptoms, the MIRA worker gave the individual a sputum cup with instructions on how to produce a sputum sample. A member of ZMQ’s sputum collection team would then visit that house to transport the sample to an NTP laboratory. The samples were tested by smear microscopy, and the results were shared with ZMQ staff who subsequently shared the information with the individual with presumptive TB who then was linked with the TB center to initiate treatment if necessary.

### 2.4. C. Evaluation of Activities (Process, Outcome, and Impact Evaluation of the Intervention)

Evaluation of the intervention consisted of three different components: (1) a process evaluation to assess the uptake (reach) of the stories as well as satisfaction with the format and content, (2) an outcome evaluation to assess the TB knowledge gained from listening to the stories and (3) an impact evaluation to assess the effect of the storytelling and ACF intervention on TB case notifications in Punhana block. All data were analyzed using R Studio 4.0.2.

### 2.5. Process Evaluation for Digital Storytelling

Process evaluation included assessing the audience reached by the intervention and their satisfaction with the stories. MIRA workers documented the number of persons reached in their house-to-house and community sessions. To assess satisfaction, assuming a target population of 20,000 adolescent girls and women in Punhana block, a confidence interval of 95%, and a 5% margin of error, a sample size of 377 was calculated. Assuming a nonresponse rate of 10%, a total sample size of 415 was calculated for a reliable interpretation of the satisfaction questionnaires. The paper-based satisfaction questionnaires consisting of a 4-point Likert scale (“totally agree”, “agree”, “disagree”, and “totally disagree”) were administered by the MIRA workers to individuals encountered at house-to-house screenings as well as every attendee of the community group sessions. Questionnaires slightly differed by setting. Descriptive analysis was performed on the characteristics of all participants who completed the satisfaction questionnaire. Data were disaggregated by age, sex, and topic of story. Frequencies for each question on the satisfaction questionnaire were tabulated to display the responses.

### 2.6. Outcome Evaluation for Digital Storytelling

To determine if the stories achieved their intended goal of increasing TB awareness and knowledge, each participant in both the house-to-house and community group settings completed a TB knowledge assessment consisting of five questions on the digital platform before and after they were shown a story. In the house-to-house intervention, the user was allowed to pick the multiple-choice answer themselves, whereas in the group intervention, a consensus was reached before an answer was selected on the digital platform.

The sample size calculation for the outcome evaluation was identical to that for the process evaluation (n = 415). Mean scores for the knowledge pre- and post assessments were calculated and subsequently disaggregated by age, sex, and topic of story. Multivariate linear regression was performed to control for these variables and investigate whether the change in mean score after intervention was statistically significant. The code utilized in R Studio was as follows: lm(Scores ~ Test date + Group/Individual setting + Story Topic + Age Group + Sex).

### 2.7. Impact Evaluation of Combined Digital Storytelling and ACF Activities

To assess the impact of the digital storytelling with the ACF activities, the established TB REACH monitoring and evaluation framework was used [11]. In this framework, TB notifications are compared between the EP (where the intervention took place Punhana block) and a control population (CP), using historical notification data and notification data during the intervention timeframe. For the CP, the Nuh block in Nuh was selected because it closely matched the EP’s population size and demographic characteristics and had similar baseline TB notification data. The historical notification data were gathered for a 3-year baseline of January 2017 to December 2019 for both the EP and the CP. Since baseline data are presented by quarters and the implementation period lasted for 2 quarters longer than the observed historical baseline (Q1-4 2019), historical case notification was adjusted by adding two additional quarters from the set baseline to correct for the discrepancy. Prior to the implementation of the intervention, targets for the TB care cascade (screening, diagnosis, and treatment) were established based on the assessment of the EP baseline data. During implementation, these care cascade process indicators were recorded prospectively. Overall case notification data for the intervention period were also gathered from the National TB Elimination Program team. All data were captured using mobile tools and stored on ZMQ’s secure server only accessible by project staff. All the data were validated through monthly comparison of NTP data.

For analysis, collected process indicators were tabulated and disaggregated by sex and screening strategy (community-based or house-to-house). Second, notification data in the EP were compared to historical baseline notification data to compute how many additional cases were detected during the implementation compared to the historical baseline. Notification data from the CP were also compared to the historical baseline to provide a benchmark for comparison during the implementation period (double difference).

## 3. Results

This project resulted in the development of six digital TB stories used during TB awareness and ACF activities carried out by 50 MIRAs. The MIRAs conducted a total of 250 house-to-house screenings and 85 community group sessions in the Punhana block of Nuh.

### 3.1. Characteristics of Participants Who Completed Satisfaction and TB Knowledge Pre- and Postassessment Questionnaires

Table 1 describes the participants in each intervention, disaggregated by sex, age group, and theme of story that they were shown. In total, there were 218 participants in the house-to-house strategy and 513 participants among 72 groups in the community-based strategy. In the former, 91.3% of the participants were women and 8.7% of the participants were men. In the latter, 438 (85.4%) were women and 75 (14.6%) were men. Overall, most participants were female (637, 87.1%). The ages were similarly distributed for the house-to-house and community-based strategies. The selection of story topics was evenly distributed for the community-based strategy, while for the house-to-house strategy the most common story topic was TB Signs and Symptoms and the least common was TB Prevention.

### 3.2. Process Evaluation Results

MIRAs showed the digital stories to 19,345 persons. Of these, a total of 731 participants (513 participants in the community setting and 218 in house-to-house screenings) responded to the questionnaire. Table 2 outlines responses to the satisfaction questionnaire in both the community group settings and house-to-house screening. Overall satisfaction ratings for the stories were favorable with the majority of participants either indicating “totally agree” or “agree” in response to most of the statements. While, overall, most responses were favorable for the stories, a small portion of participants responded negatively to some of the statements with 13% (n = 96) finding the story length inappropriate, 3% (n = 22) reporting not liking the comic characters, and 4% (n = 8) in the house-to-house screenings indicating not liking the digital format. No significant differences were found between participants’ satisfaction with the stories based on the intervention setting.

### 3.3. Outcome Evaluation Results

Table 3 shows the difference in TB knowledge assessment scores before and after the viewing of the TB stories. Participants could have scored anywhere between 1 and 5 points. The mean score after the viewing showed a significant change of 1.37 points (*p* < 0.05) compared to the mean score before the viewing. After stratifying scores across study variables, overall change was highest for young participants, aged under 17 years. For all variables, mean test scores increased after the viewing of the YST TB stories.

A multivariate regression (Table 4) displays that the mean score after the viewing still shows a significant increase in score, even after controlling for age, sex, story topic, and screening strategy. Additionally, the mean score increased by 0.91 (*p* < 0.01) points when the test was completed in a house-to-house setting (individual) rather than when the group had to reach a consensus in the community-based strategy, controlling for sex, age, story topic, and viewing of the story. Scores for signs and symptoms and TB treatment showed the highest significant changes before and after the viewing of the YST TB stories.

### 3.4. Impact Evaluation

Table 5 outlines process indicator data for the ACF intervention disaggregated by sex and the screening strategy. MIRAs screened a total of 19,345 individuals, of whom 2178 (11.3%) were people with presumptive TB; of those, 1566 (72%) were tested for TB. The proportion of people tested (72%) was marginally lower than expected (the target established at baseline was 80% testing). The project assumption ahead of implementation was to at least test 80% of identified presumptive cases. Under field conditions in active case-finding interventions, the loss expected was around 15–20% based on anecdotal evidence. Following implementation, it was observed that 72% of identified presumptive were tested against set 80% testing targets. It should be noted that the implementation also coincided with the first wave of COVID-19 observed during Q2-3 2020 and the huge second wave observed in Q2 2021. Consequently, COVID-19-imposed lockdowns, access barriers to health facilities, and preferences to access private health care, among others, were the reasons behind the suboptimal testing of identified presumptive cases against set targets (72% vs. baseline target of 80%).

A total of 255 (11.7%) individuals were diagnosed with all forms (AF) of TB (either through bacteriological testing, clinical examination, or with extrapulmonary TB), of which 186 (72.9%) had bacteriologically confirmed TB (Bac+ TB). Of those diagnosed, almost all, 253 (99.2%), initiated treatment. More individuals were screened and diagnosed with TB through the house-to-house intervention, and more women than men were screened through this intervention as well. The overall project yield by end of June 2021 was 185 Bac+ TB and 253 AF cases, resulting in a case-finding target attainment of 102% for Bac+ and 69% for AF cases (proposed yield targets at baseline = 182 Bac+ and 365 AF cases).

The combined digital story and ACF intervention resulted in increased case finding in the EP when compared to the CP as evidenced in Table 6. Compared to historical baseline (corrected by adding 2 additional quarters from the baseline year to account for the additional 2 quarters of implementation).

The unadjusted additionality is positive at 175 Bac+ and negative at −65 for AF cases. While detection of AF of TB decreased in both blocks, the proportion of decrease was lower in Punhana block (as demonstrated by significant double difference that was 79% for Bac+ and 31% for AF) cases. It remains reasonable to apply this double difference (difference in % change from baseline between EP and CP) given the percentage change from baseline witnessed a significant decline of −23% for B+ and −39% for AF in CP, while EP had gained 56% for B+ and observed a significantly lower decline of −8% for AF cases. The additional notification by yield ratio at 0.95 for Bac+ suggests the project contributions to the overall additionality observed in the evaluation population. It should be noted that COVID-19 impacted implementation during Q2–3 2020 and Q2 2021.

## 4. Discussion

ZMQ implemented an ACF intervention using an innovative digital storytelling approach to raise TB awareness and influence residents in the Punhana block community to undergo TB screening through two different strategies. This approach to ACF lead to increased knowledge and awareness about TB and resulted in an increase in TB case notifications in Punhana block.

Storytelling to create awareness and educate the community about TB was an important component of this intervention. While storytelling has been used to support various other public health initiatives, limited use with TB ACF has been documented [12]. This technique can be successful for sharing facts on a topic and can also be used to influence behavior change. For example, a recent review by Woudstra and Suurmond (2019) examined how storytelling can influence the decision to undergo colorectal cancer screening [13]. This review found that narratives with educational messages could influence decision making through addressing perceived barriers and increasing self-efficacy by depicting experiences from similar individuals and focusing on the positive outcomes of being screened and/or the negative outcomes of not being screened.

A key component of the ZMQ stories was the Story Labs that used a community participatory approach to develop the story characters and themes. In the Story Labs, the community participants were engaged to help develop stories that reflected their own experiences while incorporating educational TB messages. The participatory approach is rooted in classic community education and empowerment methods as espoused by the Brazilian adult educator Paulo Freire [14]. The participatory approach allows community members to recognize themselves in the stories being told, which can ultimately empower and enable them to make decisions to improve their health [15]. The benefits of participatory approaches for community development and health have been well documented. For example, a recent study in rural Benin highlighted using storytelling and a participatory approach to encourage the education of girls. The method allowed community members to reflect upon their own lives to tell their own stories and influence other community members to overcome barriers to educating girls. In addition to the Story Labs, another unique feature was the digital format for the stories. This format allowed for community members with low literacy to listen to and watch the stories and receive consistent and accurate messages about TB prevention and treatment. For this intervention, women were purposefully overrepresented to provide them a voice and empower them in health decision making using an accessible format for those with low literacy skills [5].

The success of storytelling intervention is supported by overall high satisfaction expressed by participants as well as the TB knowledge gained by the audience before and after listening to the stories. Very few individuals indicated dissatisfaction with some aspects of the storytelling, particularly regarding whether storytelling helped them understand TB or the story, length of stories, use of comic characters, relevance of the topics, and reluctance to share stories with friends or family, which are factors that should potentially be further explored and improved.

It is possible that some agreements with the statements in the satisfaction questionnaire are due to response bias since questionnaires were administered by MIRAs who have an important role in the health of the community. Administration of the questionnaire by MIRAs was necessary due to the low literacy of the population. However, the increase in TB knowledge (measured by increases in the knowledge assessment scores) after viewing of stories further demonstrates the usefulness of the digital stories in increasing TB education and awareness.

The ACF using the storytelling resulted in 19,345 individuals screened for TB and 255 people diagnosed with TB. It is important to note that this intervention took place against the backdrop of the COVID-19 pandemic which eventually resulted in a country-wide lockdown in India from March to April 2020 and April to June 2021. These lockdowns resulted in an overall disruption of TB service delivery leading to reported TB notifications remaining below the prepandemic average, despite predicted rises in TB infection and mortality [16,17]. Due to the challenging and unprecedented context, case notification decreased in both the CP and EP. Despite this, decreases in TB notifications were much higher in the Punhana block, particularly when looking at individuals with Bac+ TB. When compared to the historical baseline, higher case notification during the implementation period was observed in Punhana block (EP), resulting in an increase in Bac+ TB notifications with 56% change from baseline and 8% decrease for AF, compared to a 23% decrease in Bac+ TB and a 39% decrease in AF TB in Nuh. This may be due to the intervention providing transport of sputum samples, making TB testing more accessible during times when individuals could not access other TB services. Moreover, the pandemic may help explain why more individuals accessed the intervention through house-to-house screening and not through the community-based strategy given that there may have been more difficulty accessing community settings, or a reluctance to do so due to a fear of infection. Both strategies resulted in high treatment initiation (99.2%). Given these results, it is possible that the implementation of the ACF activities helped offset the effects of the pandemic and related lockdowns.

*Lessons learned* from this project included the importance of using a systematic process to design, create, pilot, disseminate, and evaluate the digital stories. Using this process helped ensure that the stories were technically accurate and relevant for the audience. However, this process was human-resource-intensive, but collaborating with existing health workers in the area helped reduce the need for extensive training in community engagement and TB screening and treatment. Further, because impact evaluation of this intervention was based on programmatic case notification data, and not conducted through a controlled study, direct causality between using the digital stories as part of the ACF and increase in case notification cannot be established. However, based on negative additionality in the CP (Nuh block) and significant double difference, it is plausible to suggest that the interventions implemented by the project have contributed to the TB case finding and gains as reported in the EP (Punhana block).

Future studies should look at comparing similar ACF activities with and without the storytelling feature to further determine the impact that the stories can have on TB case notifications.

## 5. Conclusions

This intervention represents an innovative approach to TB case finding and education that employed a participatory, community-based approach. Digital storytelling to provide TB education was a main component of this intervention which resulted in overall high satisfaction with story content and improvements in knowledge on the presented topics. This subsequently led to increases in case notification in the context of an unprecedented and challenging pandemic and country-wide lockdowns. Given the success of this intervention, it would be beneficial to further pilot digital storytelling in different populations to better understand how these interventions can be better leveraged for TB case finding. The sustained flow of TB knowledge and communication using a digital strategy has a lasting impact on communities, thereby leading to self-active case reporting which reduces the system burden of conducting intermittent ACF activities. Furthermore, during conditions/scenarios such as a pandemic or a population living in a remote setting; it is highly impactful to provide knowledge-building and self-screening tools for improving case finding. Moreover, digital storytelling can also be used to educate and create awareness for other global health initiatives. The tools are feasible to use with low-literacy populations and help ensure that community health workers provide consistent and accurate health messages.

## Figures and Tables

**Table 1 tropicalmed-07-00049-t001:** Participant characteristics of the storyteller and TB ACF intervention by setting.

	Group Intervention	Single Intervention	Overall	Community	House-to-House	Overall
	(n = 513)	(n = 218)	(n = 731)	(n = 513) (72 Groups)	(n = 218)	(n = 731)
**Sex**						
Female	438 (85.4%)	199 (91.3%)	637 (87.1%)	438 (85.4%)	199 (91.3%)	637 (87.1%)
Male	75 (14.6%)	19 (8.7%)	94 (12.9%)	75 (14.6%)	19 (8.7%)	94 (12.9%)
**Age Group**						
Below 17 years	31 (6.0%)	19 (8.7%)	50 (6.8%)	5 (1%)	19 (8.7%)	24 (8.3%)
Between 17 and 34 years	308 (60.0%)	131 (60.1%)	439 (60.1%)	43 (59.7%)	131 (60.1%)	174 (60.0%)
Above 34 years	174 (33.9%)	68 (31.2%)	242 (33.1%)	24 (33.3%)	68 (31.2%)	92 (31.7%)
**Story Topic**						
Causes of TB	82 (16.0%)	40 (18.3%)	122 (16.7%)	11 (15.3%)	40 (18.3%)	51 (17.6%)
DR-TB	94 (18.3%)	15 (6.9%)	109 (14.9%)	13 (18.1%)	15 (6.9%)	28 (9.7%)
Prevention	94 (18.3%)	11 (5.0%)	105 (14.4%)	12 (16.7%)	11 (5.0%)	23 (7.9%)
Signs and Symptoms	90 (17.5%)	61 (28.0%)	151 (20.7%)	13 (18.1%)	61 (28.0%)	74 (25.5%)
Stigma and Discrimination	73 (14.2%)	42 (19.3%)	115 (15.7%)	11 (15.3%)	42 (19.3%)	52 (18.3%)
TB Treatment	80 (15.6%)	49 (22.5%)	129 (17.6%)	12 (16.7%)	49 (22.5%)	61 (21.0%)

Note: TB = tuberculosis; DR-TB = drug-resistant tuberculosis; ACF = active case finding.

**Table 2 tropicalmed-07-00049-t002:** Satisfaction with TB stories by intervention setting.

Question */Response **	Community n = 513	House-to-House n = 218	Overall n = 731
**I liked the story**			
Totally agree	374 (72.9%)	154 (70.6%)	528 (72.2%)
Agree	139 (27.1%)	63 (28.9%)	202 (27.6%)
**This story helped me understand TB**			
Totally agree	366 (71.3%)	147 (67.4%)	513 (70.2%)
Agree	141 (27.5%)	71 (32.6%)	212 (29%)
Disagree	6 (1.2%)	0 (0%)	6 (<1%)
**I thought the story length was appropriate**			
Totally agree	130 (25.3%)	75 (34.4%)	205 (28.0%)
Agree	313 (61.0%)	112 (51.4%)	425 (58.1%)
Disagree	65 (12.7%)	31 (14.2%)	96 (13.1%)
Do not want to answer	5 (1.0%)	0 (0%)	5 (<1%)
**I understood the story**			
Totally agree	268 (52.2%)	124 (56.9%)	392 (53.6%)
Agree	240 (46.8%)	93 (42.7%)	333 (45.6%)
Disagree	5 (1.0%)	0 (0%)	5 (<1%)
Do not want to answer	0 (0%)	1 (<1%)	1 (<1%)
**I enjoyed the comic characters**			
Totally agree	199 (38.8%)	74 (33.9%)	273 (37.3%)
Agree	295 (57.5%)	141 (64.7%)	436 (59.6%)
Disagree	19 (3.7%)	3 (1.4%)	22 (3.0%)
**The story was relevant to me**			
Totally agree	278 (54.2%)	125 (57.3%)	403 (55.1%)
Agree	230 (44.8%)	92 (42.2%)	322 (44.0%)
Disagree	5 (1.0%)	0 (0%)	5(<1%)
Do not want to answer	0 (0%)	1 (<1%)	1(<1%)
**I will show this story to my friends and family**			
Totally agree	197 (38.4%)	85 (39.0%)	282 (38.6%)
Agree	313 (61.0%)	131 (60.1%)	444 (60.7%)
Disagree	3 (<1%)	2 (<1%)	5(<1%)
**I liked having the MIRA show the story to me **/I liked the digital format (video) *****			
Totally agree	295 (57.5%)	81 (37.2%)	- ^^^
Agree	218 (42.5%)	129 (59.2%)	- ^^^
Disagree	0 (0%)	8 (3.7%)	- ^^^

***** Possible responses: totally agree, agree, disagree, totally disagree, do not want to respond; ****** question only asked of community group; ******* question only asked for house-to-house screening; **^** overall total not calculated as questions differed.

**Table 3 tropicalmed-07-00049-t003:** Scores for TB knowledge assessment before and after viewing TB stories.

	Before	After	
Overall Scores			Difference (95% CI)
Mean (SD)	2.05 (1.41)	3.87 (1.34)	1.81 (1.70,1.93)
**Disaggregated Mean Scores**			
Sex			**Difference (SD)**
Female	2.07 (1.43)	3.88 (1.33)	1.81 (1.53)
Male	1.95 (1.31)	3.78 (1.45)	1.83 (1.75)
**Age Group**			
Below 17 years	2.43 (1.27)	4.67 (0.75)	2.23 (1.41)
Between 17 and 34 years	1.94 (1.38)	3.93 (1.22)	1.99 (1.44)
Above 34 years	2.18 (1.49)	3.58 (1.56)	1.40 (1.70)
**Story Topic**			
Causes of TB	1.84 (1.42)	3.96 (1.16)	1.12 (1.73)
DR-TB	1.61 (1.20)	3.17 (1.50)	1.56 (1.42)
Prevention	1.76 (0.76)	3.78 (1.64)	2.02 (1.36)
Signs and Symptoms	2.84 (1.49)	4.64 (0.69)	1.80 (1.38)
Stigma and Discrimination	1.91 (1.50)	3.45 (1.16)	1.54 (1.68)
TB Treatment	2.08 (1.47)	3.91 (1.38)	1.83 (1.66)
Intervention Strategy			
Community	1.63 (1.12)	3.73 (1.41)	2.10 (1.50)
House-to-House	3.05 (1.52)	4.20 (1.10)	1.15 (1.49)

Note: TB = tuberculosis; SD = standard deviation; DR-TB = drug-resistant tuberculosis.

**Table 4 tropicalmed-07-00049-t004:** Multivariate regression of mean scores for TB knowledge assessment before and after digital story intervention.

Scores				
Predictors		Estimates	CI	p
(Intercept)		2.20	1.29–2.28	<0.001
Before Intervention	ref			
After Intervention		1.81	1.17–1.58	<0.001
Community strategy				
House-to-house strategy		0.83	0.60–1.22	<0.001
Story topic (Causes of TB)	ref	0		
Story topic (DR-TB)		−0.36	−0.45–0.38	0.872
Story topic (Prevention)		0.06	−0.46–0.44	0.970
Story topic (Signs and Symptoms)		0.75	0.77–1.41	<0.001
Story topic (Stigma and Discrimination)		−0.26	−0.34–0.37	0.923
Story topic (TB Treatment)		0.04	0.02–0.69	0.040
Age group (Below 17 years)	ref	0		
Age group (Between 17 and 34 years)		−0.51	−0.52–0.26	0.514
Age group (Above 34 years)		−0.48	−0.61–0.21	0.348
Sex (Female)	ref	0		
Sex (Male)		−0.01	−0.45–0.41	0.923
Observations		1452		
R2/R2 adjusted		0.429/0.425		

Note: TB = tuberculosis; DR-TB = drug-resistant tuberculosis.

**Table 5 tropicalmed-07-00049-t005:** Outcome of active case-finding activities in Punhana block.

	House-to-House		Community Screening		Total
Process Indicator	Male	Female	Male	Female	n (%)
	n (%)	n (%)	n (%)	n (%)	
**Number screened**	5619	8423	2751	2552	19,345
**Number with presumptive TB (% of screened)**	657 (11.7)	893 (10.6)	289 (10.5)	339 (13.3)	2178 (11.3)
**Number tested (% with presumptive TB)**	475 (72.2)	675 (75.6)	204 (70.6)	212 (62.5)	1566 (71.9)
**Number diagnosed with Bac+ TB (% of tested)**	58 (12.2)	62 (9.2)	36 (12.5)	30 (14.2)	186 (11.9)
**Number diagnosed with AF TB (% with presumptive TB)**	80 (12.1)	82 (9.2)	53 (18.3)	40 (11.8)	255 (11.7)
**Number with Bac+ TB started on treatment (% of diagnosed with Bac+ TB)**	58 (100.0)	61 (98.4)	36 (100.0)	30 (100.0)	185 (99.5)
**Number with AF TB started on treatment (% with diagnosed AF TB)**	78 (97.5)	82 (100.0)	53 (100.0)	40 (100.0)	253 (99.2)

Note: TB = tuberculosis; Bac+ = bacteriologically confirmed TB; AF = all forms of tuberculosis including bacteriologically and clinically confirmed pulmonary TB and extrapulmonary TB.

**Table 6 tropicalmed-07-00049-t006:** Comparison between TB case notification during historical baseline and implementation period in Punhana (EP) and Nuh (CP) blocks.

Implementation Period	Quarter	Bac TB+		AF TB	
		Punhana	Nuh	Punhana	Nuh
**Historical Baseline**	January–March 2019	31	87	101	195
	April–June 2019	68	96	157	224
	July–September 2019	70	72	171	217
	October–December 2019	46	88	117	187
	Total	215	343	546	823
	**Correction factor (1.5)**	**323**	**515**	**819**	**1235**
**Implementation period**	January–March 2020	100	97	182	201
	April–June 2020	62	52	92	90
	July–September 2020	80	56	118	104
	October–December 2020	87	62	112	103
	January–March 2021	89	69	136	135
	April–June 2021	71	67	99	126
	Total	489	403	739	759
**Additional cases**		175	−123	−65	−483
**% Change from baseline**		56%	−23%	−8%	−39%
**Double difference**		79%		31%	
**Additional notifications/yield ratio**		0.95		−0.26	

## Data Availability

Tuberculosis data are subject to NTP data sharing policy and other data may be available upon request.

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
