# Peer review of "Digital Storytelling and Community Engagement to Find Missing TB Cases in Rural Nuh, India"

_tropicalmed, 2022, doi:10.3390/tropicalmed7030049_

Round 1
Reviewer 1 Report
Introduction:
Highlight the importance of TB active case finding to increase case notification rates.
Discuss research work already done on active case finding in the country and region.
Describe clearly the general and specific objectives of the study.
Methods
Method section is weak, requires improvement.
- Explain study design
- When the study was conducted?
- Where study was conducted? describe the study site.
- How stakeholders were selected and invited to participate in the story lab session.
- Which sampling method was used? (random or convenience sampling method)
-
Active TB case finding:
- Home to home screening - how homes were selected?
Community-based screening:
- How were community groups selected and approached?
Ethical concern/IRB approval
Ethical concerns and IRB approval are not mentioned.
Results:
- Link results with clearly defined objectives.
- Only 71.9% of total TB presumptive cases were tested.
- Why 28% of presumptive cases could not be tested? (It might have introduced bias in the study)
Discussion:
- Discuss the important findings of the study. (When the intervention was implemented and which approaches were employed, are parts of the methods section).
- Discuss the interpretation of results with reference to work done on the topic by other researchers.
Conclusions:
- The conclusion should help the reader to understand, why your research matter to them.
- Restate the research problem you have addressed.
- Summarize your overall findings.
- Conclude your thoughts.
Author Response
Responses for REVIEWERs
Dear Reviewers,
Thank you for the careful review and the feedback to help make our publication stronger.
We have addressed your comments as indicated below and within the attached manuscript. We have included the line numbers for where we have made the changes in the manuscript using track changes.
Thank you so much for your review and consideration. We hope our clarifications are satisfactory and address your concerns.
ZMQ
REVIEWER 1: Comments and Suggestions for Authors
Below our suggestions
- Introduction
- Highlight the importance of TB active case finding to increase case notification rates.
In paragraph 1 and 2 of introduction, we have made changes and add more information about TB ACF and notification rate.
- Discuss research work already done on active case finding in the country and region.
In paragraph 3 of introduction, we have added the information.
- Describe clearly the general and specific objectives of the study.
Mentioned in the last paragraph of introduction
- Methods
- Explain study design
We added in paragraph 1 of methods.
- When the study was conducted?
We added in paragraph 1 of methods.
- Where study was conducted? describe the study site.
We added in paragraph 1 of methods and also study site was described in paragraphs 3 and 4 of introduction.
- How stakeholders were selected and invited to participate in the story lab session.
We added in paragraph 4 of methods and also highlighted in the same paragraph about the process of invitation.
- Which sampling method was used? (random or convenience sampling method)
We added in paragraph 1 of methods.
- Active TB case finding: Home to home screening - how homes were selected?
We added in paragraph 8 of methods and also highlighted in the same paragraph about the process.
- Community-based screening: How were community groups selected and approached?
We added in paragraph 10 of methods and also highlighted in the same paragraph about the process.
- Ethical concern/IRB approval
- Ethical concerns and IRB approval are not mentioned.
We have added the information on page no. 14
- Results:
- Link results with clearly defined objectives.
The results are separated by sub-headings that relate to the same sub-headings used in the methods with very similar terminology. Please provide more insight if above-mentioned information is not clear.
- Only 71.9% of total TB presumptive cases were tested. Why 28% of presumptive cases could not be tested? (It might have introduced bias in the study)
We have added the reasons of gap (presumptive case that could not be tested) in paragraph 8 of results.
- Discussion:
- Discuss the important findings of the study. (When the intervention was implemented and which approaches were employed, are parts of the methods section).
- Discuss the interpretation of results with reference to work done on the topic by other researchers.
We have addressed both suggestions in discussion.
- Conclusions:
- The conclusion should help the reader to understand, why your research matter to them.
- Restate the research problem you have addressed.
- Summarize your overall findings.
- Conclude your thoughts.
We have addressed all suggestions in Conclusions.
Please note: All revisions made in the manuscript are marked up using the “Track Changes” function.

Reviewer 2 Report
I read with interest the paper. I find it well wrote and with good idea research
below my suggestions
- Introduction: Updata TB data on TB report 2021. Furthermore, add the impact on SARS CoV2 on diagnostic delay and increase of clinical severity of newly tb diagnosis (see and cite Increase in Tuberculosis Diagnostic Delay during First Wave of the COVID-19 Pandemic: Data from an Italian Infectious Disease Referral Hospital. Antibiotics (Basel). 2021 Mar 8;10(3):272)
- methods and results: are clear
- Discussion: interesting the role of storytelling. In my opinion it is a new aspect and new tools that we can use for tb control. Explain better also the role and involvement of community and also give some global health proposal that came from your study.
- Furthermore, add the central role of education to prevent tuberculosis and this new aspect of digital education to fight infectious diseases and also can be use on other non communicable diseases
Author Response
Responses for REVIEWERs
Dear Reviewers,
Thank you for the careful review and the feedback to help make our publication stronger.
We have addressed your comments as indicated below and within the attached manuscript. We have included the line numbers for where we have made the changes in the manuscript using track changes.
Thank you so much for your review and consideration. We hope our clarifications are satisfactory and address your concerns.
ZMQ
REVIEWER 2:Comments and Suggestions for Authors
I read with interest the paper. I find it well wrote and with good idea research
Thank you for the positive feedback
Below our suggestions
- Introduction: Updata TB data on TB report 2021. Furthermore, add the impact on SARS CoV2 on diagnostic delay and increase of clinical severity of newly tb diagnosis (see and cite Increase in Tuberculosis Diagnostic Delay during First Wave of the COVID-19 Pandemic: Data from an Italian Infectious Disease Referral Hospital. Antibiotics (Basel). 2021 Mar 8;10(3):272)
We have made this suggested change by updating the data and reference to the 2021 Global TB report and adding information on the impact of COVID on case notifications. Changes were made in paragraph 1 of introduction.
- Methods and results: are clear
Thank you we have made a few suggested changes based on Reviewer 1 comments in the attached manuscript
- Discussion: interesting the role of storytelling. In my opinion it is a new aspect and new tools that we can use for tb control. Explain better also the role and involvement of community and also give some global health proposal that came from your study.
Thank you for the positive comment. We further describe the role and involvement and engagement of the community in the methods and we have included additional information in the Discussion session in paragraph 3 of discussion. In the conclusion we provided additional information regarding the use of story telling for other global health initiatives line.
- Furthermore, add the central role of education to prevent tuberculosis and this new aspect of digital education to fight infectious diseases and also can be used on other non-communicable diseases
In discussion and Conclusion, we emphasized the importance of TB education to help influence behavior change. In the conclusion, we provided additional information regarding the use of storytelling for other global health initiatives line.
Please note: All revisions made in the manuscript are marked up using the “Track Changes” function.

Round 2
Reviewer 1 Report
|
My Comments |
Author’s Corrections |
My Comments on Revised Draft |
|
1. Introduction · Highlight the importance of TB active case finding to increase case notification rates. |
In paragraph 1 and 2 of introduction, we have made changes and add more information about TB ACF and notification rate. |
Sufficiently improved
|
|
· Discuss research work already done on active TB case finding in the country and region. |
In paragraph 3 of introduction, we have added the information. |
Needs improvement Please describe the research work done by other researchers on the topic with references If you didn’t find any published study on the topic, say clearly that no published study could be found. |
|
· Describe clearly the general and specific objectives of the study. |
Mentioned in the last paragraph of introduction |
Needs improvement Last paragraph of introduction mentions your General objective Please describe specific objectives to achieve General objective |
|
2. Methods · Explain study design |
We added in the paragraph 1 of methods.
|
Sufficiently improved |
|
· When the study was conducted?
|
We added in the paragraph 1 of methods.
|
Sufficiently improved |
|
· Where study was conducted? describe the study site. |
We added in the paragraph 1 of methods and also study site was described in paragraph 3 and 4 of introduction. |
Sufficiently improved |
|
· How stakeholders were selected and invited to participate in the story lab session. |
We added in the paragraph 4 of methods and also highlighted in the same paragraph about the process of invitation. |
Sufficiently improved |
|
- Which sampling method was used? (random or convenience sampling method) |
We added in the paragraph 1 of methods |
Sufficiently improved |
|
· Active TB case finding: Home to home screening - how homes were selected?
|
We added in the paragraph 8 of methods and also highlighted in the same paragraph about the process.
|
Needs improvement How MIRA selected homes, for T B screening. Each MIRA worker covered about 400 households, if she conducted TB screening in all 400 households? |
|
· Community-based screening: How were community groups selected and approached?
|
We added in the paragraph 10 of methods and also highlighted in the same paragraph about the process.
|
Sufficiently improved |
|
3. Ethical concern/IRB approval Ethical concerns and IRB approval are not mentioned.
|
We have added the information on page no. 14 |
Needs improvement Your answer ethical approval is not required is not acceptable. Every research proposal involving human subjects needs ethical review and approval by Institutional Review Board / Ethical Review Committee. |
|
4. Results: · Link results with clearly defined objectives.
|
The results are separated by sub-headings which relate to the same sub-headings used in the methods with very similar terminology. Please provide more insight if above mentioned information is not clear. |
Sufficiently improved |
|
· Only 71.9% of total TB presumptive cases were tested. Why 28% of presumptive cases could not be tested? (It might have introduced bias in the study)
|
We have added the reasons of gap (presumptive case that could not be tested) in paragraph 8 of results. |
Needs improvement
This is weakness of the study which might have introduced bias, clearly mention in the paper the weakness of the study. |
|
1. Discussion: Discuss the important findings of the study. (When the intervention was implemented and which approaches were employed, are parts of the methods section). · Discuss the interpretation of results with reference to work done on the topic by other researchers. |
We have addressed both suggestions in discussion. |
Sufficiently improved
|
|
2. Conclusions: · The conclusion should help the reader to understand, why your research matter to them. · Restate the research problem you have addressed. · Summarize your overall findings. · Conclude your thoughts. |
We have addressed all suggestions in Conclusions. |
Sufficiently improved |

Author Response
REVIEWER 1: Comments and Suggestions for Authors
Below are suggestions
- Introduction
- Discuss research work already done on active case finding in the country and region.
In paragraph 3 of the introduction, we have added references from another paper.
- Describe clearly the general and specific objectives of the study.
In the last paragraph of the introduction, we have included the specific objective in order to achieve a general objective.
- Methods
- Active TB case finding: Home to home screening - how homes were selected?
We added in paragraph 8 of methods.
- Ethical concern/IRB approval
- Ethical concerns and IRB approval are not mentioned.
We have added the ethic statement on page no. 14
- Results:
- Only 71.9% of total TB presumptive cases were tested. Why 28% of presumptive cases could not be tested? (It might have introduced bias in the study)
We have added the information in paragraph 6 of the results (Under Impact evaluation)
Please note: All revisions made in the manuscript are marked up using the “Track Changes” function.

Round 3
Reviewer 1 Report
I have reviewed the revised manuscript and I believe the authors have responded to my suggestions and the manuscript has been sufficiently improved.
Can be accepted for publication.